# Predicting New Concept–Object Associations in Astronomy by Mining the Literature

**Jinchu Li**
School of Computer Science,
Georgia Institute of Technology
Atlanta, GA 30332, USA
jinchu.li@gatech.edu

**Yuan-Sen Ting**
Department of Astronomy,
The Ohio State University
Columbus, OH 43210, USA
ting.74@osu.edu

**Alberto Accomazzi**
Center for Astrophysics
Harvard & Smithsonian
Cambridge, MA 02138, USA
aaccomazzi@cfa.harvard.edu

**Tirthankar Ghosal**
National Center for Computational Sciences,
Oak Ridge National Laboratory
Oak Ridge, TN 37831, USA
ghosalt@ornl.gov

**Nesar Ramachandra**
Computational Science Division,
Argonne National Laboratory
Lemont, IL 60439, USA
nramachandra@anl.gov

## Abstract

We construct a concept–object knowledge graph from the entire astro-ph corpus up to July 2025, using an automated pipeline to extract named astrophysical objects from OCR-processed papers, resolve them to SIMBAD identifiers, and link them to scientific concepts annotated in the source corpus. We then ask whether the historical structure of this graph can forecast new concept–object associations before they appear in print. Because the underlying concepts are derived from clustering, semantically related concepts inevitably overlap; we address this by applying an inference-time concept-similarity smoothing step uniformly to all methods. Evaluating across four temporal cutoffs on a physically meaningful subset of concepts, we find that an implicit-feedback matrix factorization model (Alternating Least Squares; ALS) with concept smoothing outperforms the strongest neighborhood baseline (KNN based on text-embedding concept similarity) by 16.8% on NDCG@100 (0.144 vs. 0.123) and 19.8% on Recall@100 (0.175 vs. 0.146), and exceeds the best recency heuristic by 96% and 88%, respectively. These results indicate that historical literature encodes predictive structure not captured by either global heuristics or local neighborhood voting, suggesting a path toward tools that could help triage follow-up targets for costly telescope resources.

## 1 Introduction

Scientific discovery is increasingly constrained not by a lack of results, but by the difficulty of navigating a rapidly growing literature and identifying promising new connections. The idea of using the literature itself to suggest such connections predates modern machine learning: classic literature-based discovery (LBD) frames "undiscovered public knowledge" as relationships implicitly supported by disjoint literatures (Swanson, 1986; Henry & McInnes, 2017), while link prediction in evolving graphs formalizes which missing or future edges are likely to appear (Liben-Nowell & Kleinberg, 2007; Clauset et al., 2008). In many scientific settings, edges are missing not because they are unimportant, but because discovering new interactions is costly; accurate link prediction can prioritize which candidates to test (Lü & Zhou, 2011).

What has changed recently is the feasibility of building richer representations of scientific knowledge from text. Pretrained and large language models have improved extraction and representation for scientific corpora, enabling scalable pipelines for entity recognition, normalization, and relation mining (Devlin et al., 2019; Beltagy et al., 2019; Brown et al., 2020), and have been used directly for knowledge graph completion and inductive link prediction (Yao et al., 2019; Daza et al., 2021). These advances shift the bottleneck from whether we can construct literature-derived graphs to how we evaluate and use them for discovery.

Link forecasting on literature-derived graphs has accordingly become an active research direction. SemNet constructs evolving concept networks from scientific text and tests whether historical snapshots predict future connections (Krenn & Zeilinger, 2020); Science4Cast standardizes the protocol of training up to a temporal cutoff and predicting which links appear afterward (Aghajohari et al., 2021). More recently, Impact4Cast scales forecasting to tens of millions of papers (Gu & Krenn, 2024), while biomedical studies adapt LBD to time-sliced link prediction and examine how evaluation windows affect metrics and interpretation (Pu et al., 2023).

Astronomy is well suited for this task. Much of astronomical research is observational and object-centric, with progress often hinging on characterizing individual celestial objects—their physical properties, formation histories, and connections to broader phenomena. Observing time on research-grade telescopes is a scarce and expensive resource, so any tool that helps triage which objects merit follow-up has direct scientific value. As next-generation facilities such as the Vera C. Rubin Observatory begin generating vast streams of new detections, the need for such automated prioritization will only grow; the present work offers a proof of concept toward that goal. The field also benefits from well-established entity-resolution infrastructure: SIMBAD, a reference database that catalogs known astronomical objects and merges the many aliases a single object may carry (e.g., common names, catalog numbers, survey designations) into unique canonical identifiers, provides persistent identifiers and bibliographic links (Wenger et al., 2000), and ADS supports object-aware literature access through cross-linking with object databases (Henneken, 2022; Eichhorn, 1998). Yet much of the recent astronomy-facing language-model work emphasizes semantic search, entity extraction, or knowledge graph construction (Grezes et al., 2021; Sun et al., 2024), rather than temporal link prediction as a discovery mechanism.

Despite these databases, no systematically constructed concept–object knowledge graph exists in astronomy, to our knowledge. SIMBAD catalogs objects but does not map them to the scientific concepts they relate to; ADS links papers to objects but does not extract structured concept–object associations. Building such a graph requires bridging concept extraction and object resolution across the full literature.

Our work addresses this gap. We build an automated pipeline to construct what is, to our knowledge, the first large-scale concept–object knowledge graph in astronomy, resolving object mentions to SIMBAD identifiers and linking them to scientific concepts across the entire astro-ph corpus. As a proof of concept, we evaluate this graph through a temporal forecasting task: given all associations observed up to a cutoff year, we test whether a collaborative-filtering model can predict which objects will newly appear in connection with a given concept. Figure 1 provides an overview.

## 2  DATASET

### 2.1  SOURCE DATASET

Our work builds directly on a previously released, large-scale literature-mining dataset developed by Ting et al. (Ting et al., 2025). This source corpus comprises 408,590 astrophysics papers from the arXiv `astro-ph` category, spanning 1992 through July 2025, each converted to machine-readable text via an OCR pipeline.

In that dataset, Ting et al. applied an LLM-based extraction pipeline to identify key scientific and methodological concepts from each paper, yielding on average ∼10 raw concept mentions per paper. These mentions were then consolidated via K-means clustering in embedding space (using OpenAI's `text-embedding-3-large` model), producing a controlled vocabulary of 9,999 unique concepts spanning eight high-level categories (e.g., Cosmology & Nongalactic Physics, High Energy Astrophysics, Galaxy Physics, Statistics & AI). The final dataset contains approximately 3.8 million

paper–concept associations, with each concept accompanied by a descriptive text definition and a fixed embedding vector.

We treat this concept vocabulary, its associated embeddings, and the paper–concept links as fixed inputs, and focus on augmenting the dataset with a new object-extraction layer and temporal concept–object graph construction described below. We additionally use the provided concept embeddings for inference-time concept smoothing, which mitigates the semantic overlap inherent in clustering-derived concepts.

## 2.2 OBJECT EXTRACTION AND SIMBAD RESOLUTION

For each paper, we prompt GPT-5-mini with the title, abstract, and full OCR text to extract candidate astronomical object mentions intended for SIMBAD-style resolution. The extraction procedure is summarized in Appendix A; the full prompt and output schema are available in the Data and Code Availability section. For each extracted object, the model returns: (i) a single designation string, (ii) a semantic role describing how the object is used in the paper (e.g., primary subject, sample member, counterpart/host, calibration/reference), (iii) a study mode (new observations, archival/reanalysis, catalog compilation, theory/simulation, or incidental mention), and (iv) a short evidence span with its source (title/abstract/body).

Raw model outputs are cleaned and filtered to remove malformed or spurious mentions, including non-object entities (instruments, surveys, facilities), sky regions/fields, generic classes, and multi-alias strings. We validate all outputs against the fixed role/study-mode schema (and a single-designation constraint) and discard entries that fail validation. We then canonicalize and deduplicate the remaining object strings and resolve each unique name against SIMBAD to merge aliases into canonical identifiers (Wenger et al., 2000); names that cannot be resolved are discarded and their mention instances removed. Across 290,501 papers, the model produces 1,741,687 raw object mentions (396,297 unique name strings); after filtering, 1,621,446 mentions remain (376,692 unique names), of which 163,775 unique names (43.48%) successfully resolve, corresponding to 1,150,553 surviving mention instances (66.06% of raw mentions). After alias merging, we retain 100,560 unique SIMBAD objects across the corpus.

## 2.3 CONCEPT–OBJECT GRAPH CONSTRUCTION

We combine the paper–concept associations from the source dataset with the paper–object associations extracted and resolved in the previous step to form a concept–object bipartite graph. An edge between concept $c$ and object $o$ exists if and only if at least one paper mentions $o$ in connection with $c$. Each mention inherits the publication year of its source paper, enabling temporal analysis.

Not all mentions carry equal signal: an object that is the primary subject of a study based on new observations is more informative than one mentioned only in passing as a comparison point. To capture this, we assign each concept–object edge a weight that reflects both how many papers support the link and how prominently each mention figures in its paper. As described in Section 2.2, the extraction step annotates every mention with a semantic role and a study mode. Role captures how the object functions in the paper (e.g., primary scientific target vs. member of a sample vs. calibration/reference), while study mode captures the type of analysis applied (new observations, archival/reanalysis, catalog compilation, theory-only, or context-only). We map these annotations to fixed weights chosen heuristically from domain intuition—for example, a primary subject studied via new observations should receive more weight, compared to a mere calibration/reference mention in a theoretical context.

Formally, the weight of a concept–object edge is computed by aggregating per-mention strengths and applying a log transform:

$$w_{c,o} = \log\Big(1 + \sum\nolimits_{m \in \mathcal{M}(c,o)} a(m)\Big),\tag{1}$$

where $\mathcal{M}(c,o)$ is the set of paper-level mentions linking concept $c$ to object $o$. Each mention's strength is the product of a role weight and a study-mode weight:

$$a(m) = \rho_{r(m)}\, \gamma_{\sigma(m)},\tag{2}$$

where $r(m)$ and $\sigma(m)$ are the role and study mode of mention $m$, and $\rho$ and $\gamma$ are fixed lookup weights ($\rho \in [0.25, 3.0]$, $\gamma \in [0, 1.25]$) for roles and study modes respectively, listed in Appendix B.

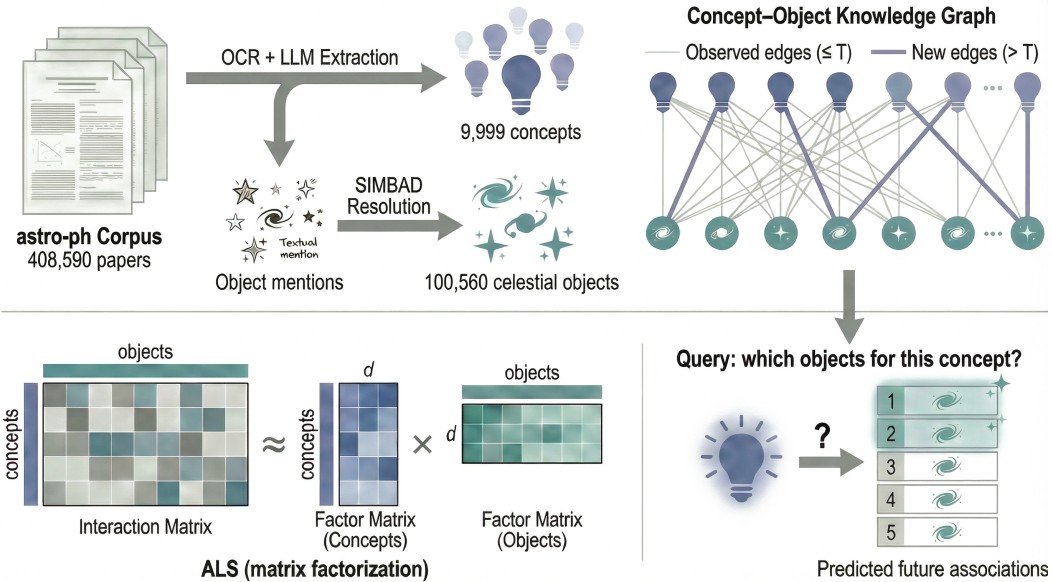

Figure 1: Pipeline overview. **Top left:** the astro-ph corpus (408,590 papers) is processed via OCR and LLM extraction to produce 9,999 concepts and raw object mentions, which are resolved through SIMBAD into 100,560 unique celestial objects. **Top right:** these are combined into a concept–object knowledge graph, split at a temporal cutoff $T$ into observed edges ($\leq T$, gray) and new edges ($> T$, purple). **Bottom left:** ALS approximates the interaction matrix as a product of low-rank concept and object factor matrices. **Bottom right:** at inference, the model ranks candidate objects for a query concept to predict future associations.

The log transform compresses the dynamic range so that edges supported by many weak mentions do not dominate edges supported by a few strong ones. We additionally record a timestamp for each concept–object pair, defined as the earliest publication year among its supporting papers; this timestamp is what enables the temporal train/test split used in our evaluation (Section 3.1). Representative edges from the concept–object graph for two illustrative concepts, including aggregated edge weights and first-appearance years, are shown in Appendix C.

## 3 METHODS

The central question is whether the concept–object graph contains enough structure to anticipate future associations. We adopt a temporal holdout protocol: freeze the graph at a cutoff year $T$, train on all associations up to that point, and evaluate predictions of associations that first appear after $T$.

### 3.1 PROBLEM DEFINITION

Let $C$ denote the set of concepts and $O$ the set of resolved astrophysical objects. From the graph $G = (C, O, E)$ constructed in Section 2, an edge $(c, o) \in E$ carries a positive weight $w_{c,o} > 0$ and a first-appearance year $y_{c,o}$.

Given a cutoff year $T$, we split the graph temporally: the training graph $G_{\leq T}$ contains only edges whose first-appearance year is at most $T$, with edge set $E_{\leq T}$, while the held-out set $E_{>T}$ contains edges appearing strictly after $T$. For each concept $c$, the task is to rank candidate objects $\mathcal{O}_c^{\mathrm{cand}} = \{o \in O \mid (c, o) \notin E_{\leq T}\}$—that is, objects not yet associated with $c$—so that the ones in $E_{>T}$ are ranked highly. For example, if the concept "High-Redshift Quasars" has been linked to a set of known quasars before $T$, the model should rank the quasars that will first appear in connection with this concept after $T$ above the remaining candidates.

## 3.2 Model and Baselines

The ranking problem defined above can be viewed as a recommendation task: given the historical concept–object associations, recommend which objects a concept is likely to be associated with next. We treat this as an implicit-feedback problem, meaning we observe which concept–object pairs have co-occurred (positive signal) but never observe explicit "negative" pairs—a concept not yet linked to an object may simply reflect an association that has not yet appeared in print, rather than one that is irrelevant. To work with standard matrix methods, we represent the graph as an interaction matrix $W \in \mathbb{R}_{\geq 0}^{|C| \times |O|}$, where $W_{c,o} = w_{c,o}$ for observed edges and $W_{c,o} = 0$ for pairs with no supporting papers. The goal is to learn a scoring function that assigns high scores to zero entries that are likely to become nonzero in the future.

**Matrix Factorization.**  Our primary model is implicit Alternating Least Squares (ALS) (Hu et al., 2008), a matrix factorization method. The core idea is that the interaction matrix $W$, despite being very large (9,999 concepts $\times$ 100,560 objects), has low-rank structure: the patterns of which concepts associate with which objects are governed by a much smaller number of latent factors (e.g., shared physical properties, observational techniques, or research themes). We approximate $W$ by representing each concept $c$ as a vector $p_c \in \mathbb{R}^d$ and each object $o$ as a vector $q_o \in \mathbb{R}^d$ (where $d \ll |C|, |O|$ is the latent dimension), such that their dot product $p_c^\top q_o$ estimates the strength of their association. Because concepts that share similar objects are pushed toward similar vectors during training, the model can generalize to predict associations for concept–object pairs not seen in the training data.

Concretely, the model minimizes a confidence-weighted squared error objective with $\ell_2$ regularization:

$$\min_{\{p_c\},\{q_o\}} \sum_{c \in C} \sum_{o \in O} (1 + \alpha w_{c,o}) \left( \mathbb{I}[w_{c,o} > 0] - p_c^\top q_o \right)^2 + \lambda \left( \sum_c \|p_c\|_2^2 + \sum_o \|q_o\|_2^2 \right), \quad (3)$$

where $\mathbb{I}[w_{c,o} > 0]$ is a binary indicator of whether an association has been observed, $\alpha = 10$ is a hyperparameter controlling how much edge weight amplifies confidence, and $(1 + \alpha w_{c,o})$ scales the confidence placed on each entry. The additive 1 ensures that even unobserved pairs (where $w_{c,o} = 0$) retain a baseline confidence of 1, so they collectively push the model toward predicting low scores by default; observed pairs override this with confidence $1 + \alpha w_{c,o} > 1$, and entries with higher edge weight are fit more tightly. The regularization strength $\lambda = 0.05$ prevents overfitting. We set the latent dimension to $d = 128$ and run 30 ALS iterations. The predicted relevance score for a concept–object pair is then $s_{\text{ALS}}(c, o) = p_c^\top q_o$.

**Concept-embedding similarity weights.**  Because the 9,999 concepts are derived from K-means clustering, semantically related concepts inevitably overlap, fragmenting associations that are scientifically coherent (see Appendix D for detailed discussion). We define a set of concept-embedding similarity weights that account for this overlap; these weights serve as a common building block for both a concept-aware KNN baseline and an inference-time smoothing step described below.

Given the fixed concept embeddings $\{e_c\}$ from the source dataset, we form a $k$-nearest-neighbor set $\mathcal{N}_k(c)$ for each concept $c$ and define nonnegative neighbor weights by clipping negative cosine similarities and $\ell_1$-normalizing:

$$S_{c,c'} = \frac{\max(\cos(e_c, e_{c'}), 0)}{\sum_{c'' \in \mathcal{N}_k(c)} \max(\cos(e_c, e_{c''}), 0)}, \quad (4)$$

for $c' \in \mathcal{N}_k(c)$, and $S_{c,c'} = 0$ otherwise. If all similarities in $\mathcal{N}_k(c)$ are non-positive, we fall back to uniform weights over $\mathcal{N}_k(c)$.

**Baselines.**  We compare ALS against several non-parametric baselines, each testing a different hypothesis about what drives future associations.

**Random** ranks candidate objects uniformly at random, assigning each object a score drawn from a uniform distribution: $s_{\text{rand}}(o) \sim \text{Uniform}(0, 1)$. This establishes a lower bound and verifies that the evaluation protocol behaves as expected.

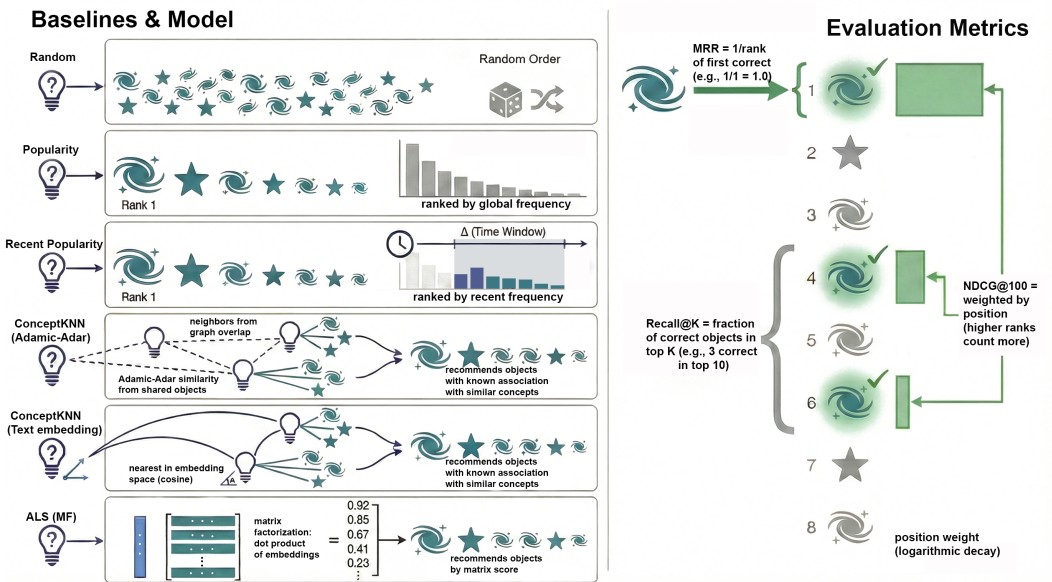

Figure 2: Visual overview of baselines and evaluation metrics. **Left:** how each method ranks candidate objects for a query concept—from top to bottom: Random (shuffled ordering), Popularity (global frequency), RecentPopularity (time-windowed frequency), ConceptKNN-AA (Adamic–Adar neighbor aggregation), ConceptKNN-TextEmb (embedding-based neighbor aggregation), and ALS (dot product of learned latent factors). **Right:** how the three evaluation metrics score a ranked list; green checkmarks denote correct held-out associations. MRR rewards placing the first correct object high, Recall@$K$ measures coverage in the top $K$, and NDCG@100 assigns position-discounted credit.

**Popularity** tests whether globally prominent objects are more likely to appear in future associations regardless of the query concept, scoring each object by its total training edge weight $s_{\mathrm{pop}}(o) = \sum_{c \in C} w_{c,o}$. The resulting ranking is concept-agnostic.

**RecentPopularity** ($\Delta \in \{3, 5\}$ years) restricts the sum to papers published in $(T - \Delta, T]$, testing whether short-term trends carry more signal than lifetime popularity. Like Popularity, it is concept-agnostic.

**ConceptKNN family.** We define a concept-aware non-parametric family of baselines that borrow evidence from similar concepts. For a query concept $c$, we form a neighbor set $\mathcal{N}_k(c)$ under a chosen concept–concept similarity and score candidate objects by

$$s_{\mathrm{knn}}(c, o) = \sum_{c' \in \mathcal{N}_k(c)} \tilde{s}(c, c')\, w_{c', o}, \tag{5}$$

where $\tilde{s}(c, c')$ are nonnegative neighbor weights normalized over $\mathcal{N}_k(c)$. We treat $k$ as a hyperparameter and report results across multiple values. We use two instantiations:

**(i) ConceptKNN_AA (graph-based).** We set similarity using the Adamic–Adar index (Adamic & Adar, 2005):

$$s_{\mathrm{AA}}(c, c') = \sum_{o \in \mathcal{O}(c) \cap \mathcal{O}(c')} \frac{1}{\log |\mathcal{C}(o)|}, \tag{6}$$

and define $\tilde{s}(c, c')$ by normalizing $s_{\mathrm{AA}}(c, c')$ over $c' \in \mathcal{N}_k^{\mathrm{AA}}(c)$.

**(ii) ConceptKNN_Emb (text-embedding-based).** We set $\tilde{s}(c, c') = S_{c, c'}$, the concept-embedding similarity weights defined in equation 4. This baseline scores objects *entirely* by aggregating the associations of semantically similar concepts.

**Training and inference.** For each cutoff $T$, we train all methods on the time-filtered graph $G_{\leq T}$, keeping the concept and object vocabularies fixed across cutoffs. The ALS model is fit on the

corresponding interaction matrix to produce latent vectors for all concepts and objects; baselines compute their scores directly from training-graph statistics. At inference time, for each concept $c$ we rank all objects in the candidate set $\mathcal{O}_c^{\text{cand}}$ by their predicted score $s(c, o)$. The same candidate masking—excluding objects already associated with $c$ in $G_{\leq T}$—is applied to all methods, ensuring a fair comparison. We select ALS hyperparameters $(d, \alpha, \lambda)$ via a small grid search on a temporally valid development setting (using cutoff $T{=}2021$ for selection), and then fix $d{=}128$, $\alpha{=}10$, and $\lambda{=}0.05$ for all reported cutoffs. For ConceptKNN baselines, we report results across $k \in \{5, 10, 25, 50, 100, 150, 200\}$. Additionally, the concept-embedding weights enable smoothing any method's predictions at inference time. Given a base score $s(c, o)$, we blend it with neighbor-aggregated scores:

$$s_{\text{smooth}}(c, o) = (1 - \beta)\, s(c, o) + \beta \sum_{c' \in \mathcal{N}_k(c)} S_{c,c'}\, s(c', o), \tag{7}$$

where $\beta$ controls the mixing strength. This preserves each method's learned structure while compensating for concept-boundary artifacts. We apply smoothing uniformly to all methods, tuning $(k, \beta)$ at $T{=}2021$ and fixing $k{=}100$, $\beta{=}0.5$ for all reported experiments. Results without smoothing are in Appendix D.

### 3.3 EVALUATION PROTOCOL

Evaluation is restricted to concepts with at least 10 training associations and at least one held-out test edge in $E_{>T}$. The first condition ensures that concepts have enough history for the model to learn from; the second ensures a meaningful prediction target.

We assess ranking quality with four standard metrics, where $C_{\text{eval}}$ is the set of eligible concepts and $E_{>T}(c)$ the held-out objects for concept $c$:

**Mean Reciprocal Rank (MRR):** the reciprocal rank of the first correct held-out object, measuring how quickly a correct result appears:

$$\text{MRR} = \frac{1}{|C_{\text{eval}}|} \sum_{c \in C_{\text{eval}}} \frac{1}{\min_{o \in E_{>T}(c)} \text{rank}_c(o)}. \tag{8}$$

**Recall@$K$** (with $K \in \{10, 100\}$): the fraction of held-out objects appearing in the top $K$ predictions:

$$\text{Recall@}K = \frac{1}{|C_{\text{eval}}|} \sum_{c \in C_{\text{eval}}} \frac{|\{o \in E_{>T}(c) : \text{rank}_c(o) \leq K\}|}{|E_{>T}(c)|}. \tag{9}$$

**NDCG@100:** a ranking-quality measure that awards more credit for correct objects placed near the top (Järvelin & Kekäläinen, 2002):

$$\text{NDCG@}100 = \frac{1}{|C_{\text{eval}}|} \sum_{c \in C_{\text{eval}}} \frac{\text{DCG@}100(c)}{\text{IDCG@}100(c)}, \quad \text{DCG@}100(c) = \sum_{i=1}^{100} \frac{\mathbb{I}[\mathcal{R}_c(i) \in E_{>T}(c)]}{\log_2(i+1)}, \tag{10}$$

where

$$\text{IDCG@}100(c) = \sum_{i=1}^{\min(|E_{>T}(c)|,100)} \frac{1}{\log_2(i+1)} \tag{11}$$

is the ideal DCG obtained by ranking all held-out objects first. All metrics are macro-averaged over concepts; no test-edge information is used during training.

## 4 RESULTS

We evaluate concept–object edge prediction across four cutoff years $T \in \{2017, 2019, 2021, 2023\}$. Depending on the cutoff, the training graph contains 2.11M–2.90M concept–object edges, and the held-out set contains 0.26M–1.08M edges. Evaluation is restricted to 7.2k–7.7k eligible concepts per cutoff (each with at least 10 training associations and at least one held-out test association), with a fixed object vocabulary of 100,560 SIMBAD-resolved objects.

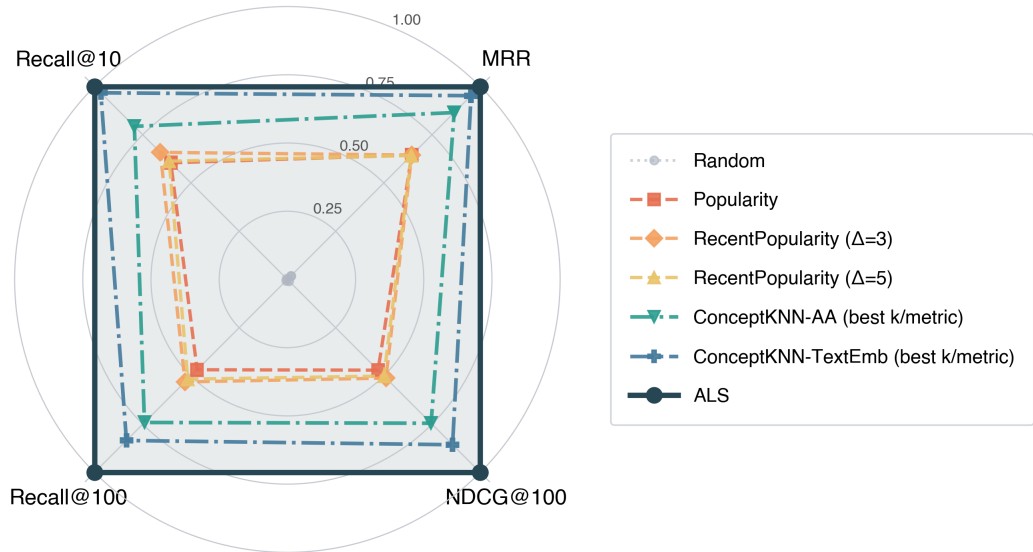

Figure 3: Radar plot comparing all methods on the physical concept subset with concept smoothing. The four axes correspond to MRR, Recall@10, Recall@100, and NDCG@100, each normalized so the best method equals 1.0. Methods shown: Random (gray dotted), Popularity (red dashed), RecentPopularity with $\Delta \in \{3, 5\}$ (orange/yellow dashed), ConceptKNN-AA (teal dashdot), ConceptKNN-TextEmb (blue dash-dot), and ALS (dark solid). ALS forms the outermost polygon, leading on all four metrics.

**(a) Absolute performance (with inference-time concept smoothing)**

| Method | MRR | Recall@10 | Recall@100 | NDCG@100 |
|---|---|---|---|---|
| Random | 0.0061±0.0020 | 0.0002±0.0001 | 0.0010±0.0001 | 0.0011±0.0003 |
| Popularity | 0.2031 | 0.0278 | 0.0817 | 0.0675 |
| RecentPopularity ($\Delta$=3) | 0.2031 | 0.0303 | 0.0927 | 0.0734 |
| RecentPopularity ($\Delta$=5) | 0.2024 | 0.0282 | 0.0902 | 0.0719 |
| ConceptKNN-AA (best $k$ per metric) | 0.2724 | 0.0365 | 0.1294 | 0.1070 |
| ConceptKNN-TextEmb (best $k$ per metric) | 0.3000 | 0.0445 | 0.1457 | 0.1230 |
| ALS | **0.3150±0.0010** | **0.0460±0.0001** | **0.1746±0.0002** | **0.1436±0.0001** |

**(b) Relative improvement of ALS over baselines (%)**

| Baseline | MRR | Recall@10 | Recall@100 | NDCG@100 |
|---|---|---|---|---|
| Random | 5054.48 | 20684.20 | 18176.33 | 12756.58 |
| Popularity | 55.13 | 65.43 | 113.75 | 112.66 |
| RecentPopularity ($\Delta$=3) | 55.11 | 51.81 | 88.31 | 95.63 |
| RecentPopularity ($\Delta$=5) | 55.62 | 62.90 | 93.60 | 99.85 |
| ConceptKNN-AA (best $k$ per metric) | 15.62 | 26.05 | 34.94 | 34.20 |
| ConceptKNN-TextEmb (best $k$ per metric) | 5.01 | 3.39 | 19.80 | 16.78 |

Table 1: Link prediction performance on the physical concept subset with concept smoothing, averaged across four cutoffs. ALS and Random are stochastic and reported as mean±std over seeds; all other baselines are deterministic. KNN baselines: best $k$ per metric. Panel (b): relative improvement of ALS over each baseline (%).

**Evaluation concept subset.** Not all concept categories are equally informative for forecasting astrophysical associations. Concepts tied to instrument usage or broadly applicable methodologies yield associations driven more by scheduling or cross-field applicability than by astrophysical structure, diluting evaluation. We therefore report headline results on a *Physical* subset that excludes Statistics & AI, Numerical Simulation, and Instrumental Design concepts (details in Appendix E).

**(a) Recall@100**

| Method | 2017 | 2019 | 2021 | 2023 |
|---|---|---|---|---|
| Random | 0.0010±0.0002 | 0.0009±0.0002 | 0.0008±0.0003 | 0.0012±0.0004 |
| Popularity | 0.0764 | 0.0812 | 0.0865 | 0.0826 |
| RecentPopularity ($\Delta$=3) | 0.0841 | 0.0846 | 0.1001 | 0.1021 |
| RecentPopularity ($\Delta$=5) | 0.0816 | 0.0845 | 0.0964 | 0.0982 |
| ConceptKNN-AA (best $k$) | 0.1201 | 0.1317 | 0.1412 | 0.1450 |
| ConceptKNN-TextEmb (best $k$) | 0.1324 | 0.1429 | 0.1547 | 0.1594 |
| ALS | 0.1463±0.0004 | 0.1596±0.0004 | 0.1766±0.0005 | 0.1851±0.0005 |

**(b) NDCG@100**

| Method | 2017 | 2019 | 2021 | 2023 |
|---|---|---|---|---|
| Random | 0.0015±0.0003 | 0.0012±0.0003 | 0.0009±0.0002 | 0.0008±0.0002 |
| Popularity | 0.0832 | 0.0744 | 0.0647 | 0.0478 |
| RecentPopularity ($\Delta$=3) | 0.0909 | 0.0771 | 0.0704 | 0.0552 |
| RecentPopularity ($\Delta$=5) | 0.0882 | 0.0775 | 0.0682 | 0.0536 |
| ConceptKNN-AA (best $k$) | 0.1369 | 0.1226 | 0.1065 | 0.0848 |
| ConceptKNN-TextEmb (best $k$) | 0.1485 | 0.1349 | 0.1195 | 0.0954 |
| ALS | 0.1617±0.0002 | 0.1459±0.0002 | 0.1306±0.0003 | 0.1070±0.0003 |

Table 2: Per-cutoff performance on the physical concept subset without smoothing. ALS leads at every cutoff on both metrics. ALS and Random: mean±std over seeds; other baselines are deterministic. KNN baselines: best $k$ per cutoff.

Table 1 and Figure 3 report performance averaged across all four cutoffs with concept smoothing applied uniformly to all methods. ALS is the strongest method on all four metrics, exceeding the best text-embedding KNN baseline by 5.0% on MRR, 3.4% on Recall@10, 19.8% on Recall@100, and 16.8% on NDCG@100. The gains over recency heuristics are substantially larger: ALS improves over the strongest recency baseline by 88% on Recall@100 and 96% on NDCG@100. The margin is widest on long-horizon retrieval metrics (Recall@100, NDCG@100)—precisely the regime most relevant to practical triaging, where a researcher scans a moderately sized candidate list. On early-rank metrics the advantage is smaller but still positive, indicating that matrix factorization captures complementary structure beyond local neighborhood voting. Results without smoothing (Appendix D) preserve the qualitative ordering but show a narrower ALS advantage on early-rank metrics; smoothing most strongly benefits ALS, consistent with the idea that learned latent factors propagate information more effectively once cluster-boundary artifacts are mitigated.

Because aggregate improvements could in principle be driven by a single temporal split, Table 2 breaks down Recall@100 and NDCG@100 by cutoff year (without smoothing, for a cleaner comparison of base methods; smoothed per-cutoff trends are in Appendix F). ALS outperforms all baselines at every cutoff on both metrics, confirming that the gains are consistent across time. RecentPopularity becomes more competitive at later cutoffs but does not close the gap; KNN baselines improve steadily with $T$ but remain behind ALS.

## 5 DISCUSSION, IMPLICATIONS, AND FUTURE DIRECTIONS

**Summary of findings.** With inference-time concept smoothing to mitigate the semantic overlap inherent in clustering-derived concepts, ALS consistently outperforms all baselines across every metric and temporal cutoff, with the largest gains on long-horizon retrieval measures most relevant to practical triaging. These improvements are concentrated in astrophysically meaningful concepts, indicating that the literature-derived graph captures predictive structure beyond local similarity or short-term trends.

**Implications for literature-driven discovery.** The broader implication is that systematically mining the published literature can surface latent structure that complements existing databases such as SIMBAD and ADS—not by cataloging objects or indexing papers, but by mapping the evolving relationship between scientific ideas and the objects they concern. Such a graph could serve as a

building block for tools that help astronomers prioritize follow-up observations, identify underexplored objects for a given research question, or flag emerging trends across subfields.

**Limits of prediction and scientific judgment.** At the same time, a predictive tool based on historical patterns carries an inherent tension. While our results show that future concept–object associations are statistically forecastable from past literature, some of the most consequential discoveries in astronomy arise precisely from unexpected associations—objects that no one predicted would be relevant to a given research question. A triaging system that ranks objects by expected relevance risks reinforcing existing research trajectories at the expense of serendipitous findings. Balancing systematic prioritization with openness to the unexpected is not a problem that any algorithm can resolve on its own; it is ultimately a matter of scientific judgment and community practice. We view the graph presented here as a tool to inform that judgment, not to replace it.

**Methodological limitations and design choices.** The concept–object graph depends on OCR quality, LLM-based extraction accuracy, and SIMBAD resolution, each of which can introduce noise. In addition, our evaluation targets are defined by first appearance in the literature rather than by a ground-truth notion of physical discovery, so predictions partly reflect evolving scientific attention and publication dynamics.

Because the 9,999 concepts are obtained by clustering raw extractions, some degree of semantic overlap is unavoidable: closely related or near-synonymous concepts may split associations that are scientifically coherent. Our inference-time concept smoothing step is a simple and transparent way to mitigate this, but more principled approaches—such as hierarchical concept representations, soft assignment, or jointly learned concept embeddings—could potentially address concept overlap more directly and are a promising direction for future work.

**Scope and future directions.** The object vocabulary is limited to the $\sim$100,560 objects mentioned by name in the literature, a small fraction of known astronomical sources, because our pipeline captures only objects that individual papers discuss explicitly. Major surveys such as Gaia and SDSS list millions to billions of sources, most of which are never referred to individually in the text. The resulting graph therefore reflects objects that have attracted individual scientific attention rather than the full census of known sources.

Future work could extend this foundation along several fronts: time-sliced concept representations that evolve with the literature, richer temporal architectures (e.g., recurrent or attention-based models over graph snapshots), stronger graph-based baselines, and integration of catalog-level source lists from large surveys to broaden the object vocabulary beyond what is mentioned by name in the literature.

The concept–object knowledge graph, including edge weights, timestamps, and concept embeddings, are available in the Data and Code Availability section.

## DATA AND CODE AVAILABILITY

All code to reproduce the object extraction, entity resolution, training and evaluation pipeline, along with (i) the SIMBAD-resolved object identifier mapping and object metadata used in this work, (ii) paper-level extracted object mentions with role and study-mode annotations, is available at: https://github.com/JinchuLi2002/astro-link-forecasting.

## ACKNOWLEDGMENTS

This work was supported in part by the U.S. National Science Foundation under Grant AST-2406729 and by a Humboldt Research Award from the Alexander von Humboldt Foundation (YST). This work also used GPT-5-mini API access provided through the NSF National Artificial Intelligence Research Resource (NAIRR) Pilot.

Work at Argonne National Laboratory was supported by the U.S. Department of Energy, Office of High Energy Physics. Argonne, a U.S. Department of Energy Office of Science Laboratory, is operated by UChicago Argonne LLC under Contract No. DE-AC02-06CH11357. NR is supported by Laboratory Directed Research and Development (LDRD) funding from Argonne National Labora-

tory, provided by the Director, Office of Science, of the U.S. Department of Energy under Contract No. DEAC02-06CH11357.

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

## A  OBJECT EXTRACTION OVERVIEW

Astrophysical object mentions are extracted from each paper using GPT-5-mini, prompted with the paper's arXiv ID, title, abstract, and full OCR text. The model is instructed to extract only named astrophysical objects suitable for SIMBAD-style resolution (e.g., standard stellar, galaxy, or source designations) and to exclude instruments, survey or catalog names, sky regions, generic object classes, and raw coordinates unless explicitly used as a named identifier.

For each extracted object, the model returns a structured JSON record containing: (i) the object name, (ii) a role label drawn from a controlled vocabulary of 11 categories (Table 3), (iii) a study-mode label drawn from 6 categories (Table 4), (iv) a short evidence span from the source text supporting the extraction, and (v) the source location (title, abstract, or body). Role and study-mode labels follow a priority tie-breaking order when multiple labels could apply. All outputs are schema-validated; entries with invalid labels, missing required fields, or malformed JSON are discarded. A single-designation constraint further rejects names containing obvious multi-designation separators (commas, slashes, semicolons, etc.). The typical yield is $\leq 25$ objects per paper, with a hard cap of 40. The exact prompt template and JSON output schema are available in the Data and Code Availability section.

## B  ROLE AND STUDY-MODE WEIGHTS

Not all object mentions in a paper are equally informative. A paper that presents new observations of a star as its primary subject conveys a much stronger scientific association between that object and the paper's topic than one that mentions the same star in passing as a calibration reference. Treating all mentions equally would dilute genuine associations with noise from incidental references. To address this, we assign each concept–object edge a weight that reflects the scientific prominence of the underlying mentions, as described in Section 2.3.

Each mention $m$ contributes a strength $a(m) = \rho_{r(m)} \gamma_{\sigma(m)}$, where $\rho_r$ is a role weight capturing how central the object is to the paper (Table 3) and $\gamma_\sigma$ is a study-mode multiplier capturing the nature of the analysis (Table 4). The role weight ranges from 3.0 for an object that is the primary subject of a study down to 0.25 for one mentioned only for calibration or comparison. The study-mode multiplier upweights new observations (1.25) and downweights theory-only mentions (0.90); a study mode of "not applicable" (e.g., an object mentioned for context with no analysis) receives zero weight, effectively discarding that mention. These numeric values were set *a priori* based on astrophysical domain intuition to reflect relative evidentiary strength, and were not tuned on held-out data.

| Role $r$ | $\rho_r$ | Description |
|---|---|---|
| Primary subject | 3.00 | Main scientific focus of the paper |
| Host or counterpart | 2.00 | Confirmed counterpart or host |
| Candidate association | 1.50 | Tentative counterpart or host |
| Foreground or lens | 1.50 | Confirmed foreground object (e.g., lens) |
| Background or lensed source | 1.50 | Background or lensed source |
| Member of sample | 1.00 | Part of an analyzed sample |
| Non-detection or upper limit | 1.00 | Searched but not detected |
| Other | 0.75 | Does not fit other categories |
| Comparison or reference | 0.25 | Used as benchmark or reference |
| Calibration | 0.25 | Calibration standard |
| Serendipitous or field source | 0.25 | Incidental field source |

Table 3: Role weights $\rho_r$ for mention strengths. Higher weights reflect greater scientific centrality.

| Study mode $\sigma$ | $\gamma_\sigma$ | Description |
|---|---|---|
| New observation | 1.25 | New observational data presented |
| Archival or reanalysis | 1.10 | Reanalysis of existing data |
| Catalog compilation | 1.00 | Catalog or survey compilation |
| Simulation or theory | 0.90 | Theoretical or simulated study |
| Unknown | 0.50 | Study mode could not be determined |
| Not applicable | 0.00 | Mentioned for context only (discarded) |

Table 4: Study-mode multipliers $\gamma_\sigma$ for mention strengths. Higher values reflect greater observational investment; "not applicable" entries are excluded.

## C  Representative Concept–Object Graph Edges

Table 5 shows representative edges from the concept–object graph for two illustrative concepts. These examples highlight the different mechanisms by which concept–object associations arise in the literature.

For "High-Redshift Quasars," the associated objects are individual quasars that have been spectroscopically confirmed at high redshift and subsequently studied across multiple papers. Each such confirmation and follow-up study contributes mentions linking the quasar to this concept, and the accumulated edge weights reflect the depth of that literature.

For "Stellar Evolution Models," the associations arise differently. The Large and Small Magellanic Clouds (LMC and SMC) are nearby satellite galaxies that serve as key testbeds for stellar evolution because they host large, well-resolved stellar populations at known distances. New photometric and spectroscopic surveys of these galaxies continually generate data against which stellar evolution models are calibrated and tested, producing a steady stream of papers connecting these objects to the concept. Similarly, Cl Melotte 25 (the Hyades) and NGC 2682 (M67) are benchmark open clusters whose well-characterized stellar populations make them recurring targets for testing stellar evolutionary tracks.

Note that the "NAME" prefix (e.g., "NAME LMC") is part of the SIMBAD identifier convention: SIMBAD uses the "NAME" prefix for objects registered under their common names rather than a catalog designation.

| Concept | SIMBAD object | $w_{c,o}$ | First year |
|---|---|---|---|
| High-Redshift Quasars | QSO B1202-074 | 4.598 | 1995 |
| | QSO J0831+5245 | 4.489 | 1998 |
| | SDSS J114816.64+525150.3 | 4.222 | 2000 |
| | QSO J1415+1129 | 4.027 | 1997 |
| Stellar Evolution Models | NAME LMC | 5.275 | 1997 |
| | NAME SMC | 4.989 | 1998 |
| | Cl Melotte 25 | 4.830 | 1994 |
| | NGC 2682 | 4.809 | 1993 |

Table 5: Representative concept–object graph edges with edge weights and first-appearance years.

## D  Concept Smoothing: Rationale and Unsmoothed Results

The 9,999 concepts in our vocabulary are produced by K-means clustering of raw LLM-extracted concept mentions in embedding space (Section 2). While clustering produces a controlled vocabulary, it also introduces an inherent fragmentation problem: semantically related or near-synonymous concepts may be assigned to separate clusters, splitting associations that are scientifically coherent. For instance, concepts such as "Stellar Nucleosynthesis" and "Chemical Enrichment Models" may share many of the same relevant objects yet end up in different clusters, so that the evidence supporting each concept's associations is diluted across cluster boundaries.

This fragmentation affects all methods, but it is particularly consequential for approaches that treat each concept independently. A method like ALS learns latent factors from the full interaction matrix and can partially share information across related concepts through the learned embeddings; however, its per-concept predictions still reflect the fragmented associations in the training data. The concept-embedding similarity weights defined in Section 3.2 address this by explicitly sharing information among semantically similar concepts.

The same weights serve two complementary roles in our framework. **ConceptKNN_Emb** uses them to score objects *entirely* by neighbor-aggregated associations—a pure non-parametric baseline that tests whether embedding similarity alone captures the structure needed for prediction. **Inference-time concept smoothing** instead *blends* any base method's own scores with neighbor-aggregated scores, preserving each method's learned structure while compensating for concept-boundary artifacts. The key distinction is that smoothing augments a model's predictions rather than replacing

them, which is why it benefits ALS most: the latent factors learned by ALS provide complementary information that local neighbor aggregation alone cannot capture, and smoothing allows this information to propagate across cluster boundaries.

To illustrate the impact of smoothing empirically, Table 6 and Figure 4 present the same comparison as the main text but without smoothing. Without smoothing, ALS remains the strongest method on long-horizon retrieval metrics (Recall@100 and NDCG@100), improving over the best text-embedding KNN baseline by 13.4% and 9.5%, respectively. However, the text-embedding KNN baseline slightly exceeds ALS on early-rank metrics (MRR by 4.4%, Recall@10 by 5.1%), reflecting a trade-off: neighborhood-based methods are competitive at placing a small number of correct associations near the top, while matrix factorization more effectively recovers a larger fraction of future associations at broader cutoffs. Concept smoothing resolves this trade-off in favor of ALS by mitigating the concept-boundary fragmentation that neighborhood methods handle implicitly through local aggregation.

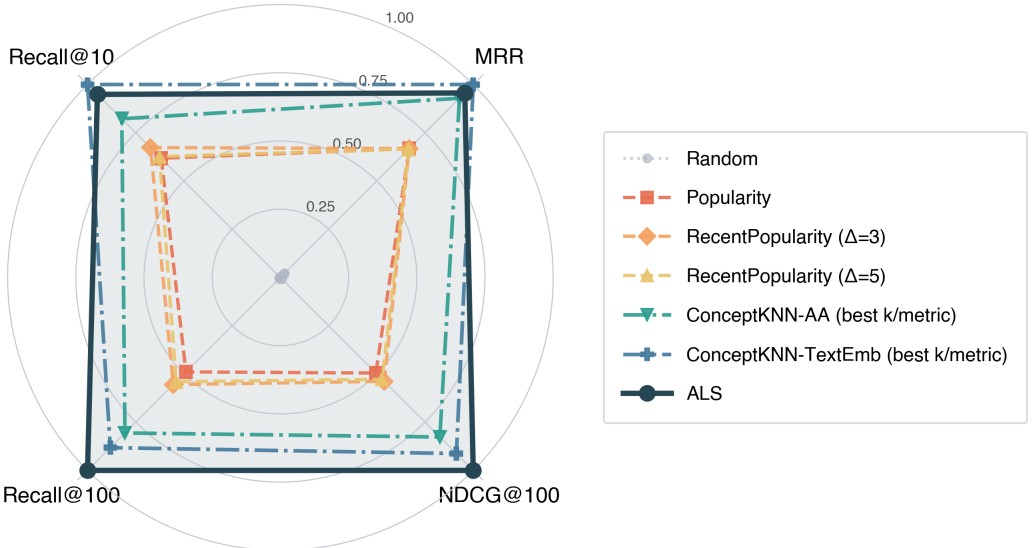

Figure 4: Radar plot on the physical concept subset *without smoothing*. ALS leads on Recall@100 and NDCG@100; ConceptKNN-TextEmb is competitive on MRR and Recall@10.

## E  CONCEPT CLASS STRATIFICATION

**(a) Absolute performance (without smoothing)**

| Method | MRR | Recall@10 | Recall@100 | NDCG@100 |
|---|---|---|---|---|
| Random | 0.0061±0.0020 | 0.0002±0.0001 | 0.0010±0.0001 | 0.0011±0.0003 |
| Popularity | 0.2031 | 0.0278 | 0.0817 | 0.0675 |
| RecentPopularity ($\Delta$=3) | 0.2031 | 0.0303 | 0.0927 | 0.0734 |
| RecentPopularity ($\Delta$=5) | 0.2024 | 0.0282 | 0.0902 | 0.0719 |
| ConceptKNN-AA (best $k$ per metric) | 0.2824 | 0.0370 | 0.1345 | 0.1127 |
| ConceptKNN-TextEmb (best $k$ per metric) | **0.3041** | **0.0450** | 0.1472 | 0.1244 |
| ALS | 0.2908±0.0011 | 0.0427±0.0001 | **0.1669±0.0002** | **0.1363±0.0002** |

**(b) Relative improvement of ALS over baselines (%)**

| Baseline | MRR | Recall@10 | Recall@100 | NDCG@100 |
|---|---|---|---|---|
| Random | 4658.16 | 19207.42 | 17370.22 | 12100.77 |
| Popularity | 43.20 | 53.68 | 104.32 | 101.81 |
| RecentPopularity ($\Delta$=3) | 43.18 | 41.02 | 80.00 | 85.65 |
| RecentPopularity ($\Delta$=5) | 43.65 | 51.33 | 85.06 | 89.66 |
| ConceptKNN-AA (best $k$ per metric) | 2.97 | 15.35 | 24.11 | 20.93 |
| ConceptKNN-TextEmb (best $k$ per metric) | −4.39 | −5.13 | 13.35 | 9.53 |

Table 6: Link prediction performance *without concept smoothing*, averaged across four cutoffs. Format follows Table 1. Bold marks the best method per metric.

The 9,999 concepts in our vocabulary span a wide range of topics, from core astrophysical phenomena (e.g., stellar evolution, galaxy mergers) to statistical methods, numerical techniques, and instrumental designs. These non-physical concepts can distort evaluation: a concept describing a specific telescope tends to link to whatever objects that telescope observes, so predicting its future associations is more about observing schedules than astrophysical insight. Similarly, a broad methodology concept (e.g., Bayesian inference) co-occurs with objects across many subfields, making its associations diffuse and uninformative. To ensure that our evaluation reflects genuinely meaningful astrophysical predictions, Section 4 stratifies concepts into subsets. This appendix specifies the exact rules used to define those subsets.

**High-level vocabulary classes.** Each of the 9,999 concepts in the vocabulary is assigned a high-level class by the source dataset. We define the "Physical" subset by including only classes that correspond to core astrophysical disciplines, and excluding classes that primarily describe tools, techniques, or simulations.

| Subset | Definition |
|---|---|
| Physical | Concepts whose high-level vocabulary class is one of: Cosmology & Nongalactic Physics; Earth & Planetary Science; Galaxy Physics; High Energy Astrophysics; Solar & Stellar Physics. |
| Excluded from Physical | Statistics & AI; Numerical Simulation; Instrumental Design. |

Table 7: Concept subsets used in stratified evaluation. The "Physical" subset retains only core astrophysical disciplines.

# F   ADDITIONAL EVALUATION PLOTS

This appendix provides supplementary plots that visualize model behavior across cutoff years and hyperparameter settings. These figures complement the aggregate tables in the main text by illustrating (i) how performance evolves over time and (ii) how neighborhood-based baselines depend on the choice of $k$. All plots are restricted to the Physical concept subset unless otherwise stated.

## F.1   PERFORMANCE TRENDS ACROSS CUTOFF YEARS

Figure 5 shows how each evaluation metric varies as a function of the cutoff year $T \in \{2017, 2019, 2021, 2023\}$. For clarity, we plot representative baselines: ALS, Popularity, the best

RecentPopularity variant, and the best-performing KNN baseline (by average performance) for each metric.

## F.2 SENSITIVITY TO NEIGHBORHOOD SIZE $k$

Figure 6 examines how KNN-based baselines depend on the neighborhood size $k$. We plot Recall@100 and NDCG@100 as functions of $k \in \{5, 10, 25, 50, 100, 150, 200\}$ for both Adamic–Adar and text-embedding KNN variants, with separate curves for each cutoff year.

Together, these plots show that (i) the relative advantages of ALS over non-parametric baselines are consistent across temporal cutoffs, and (ii) inference-time concept smoothing provides a uniform performance boost without altering qualitative trends or requiring sensitive tuning of $k$.

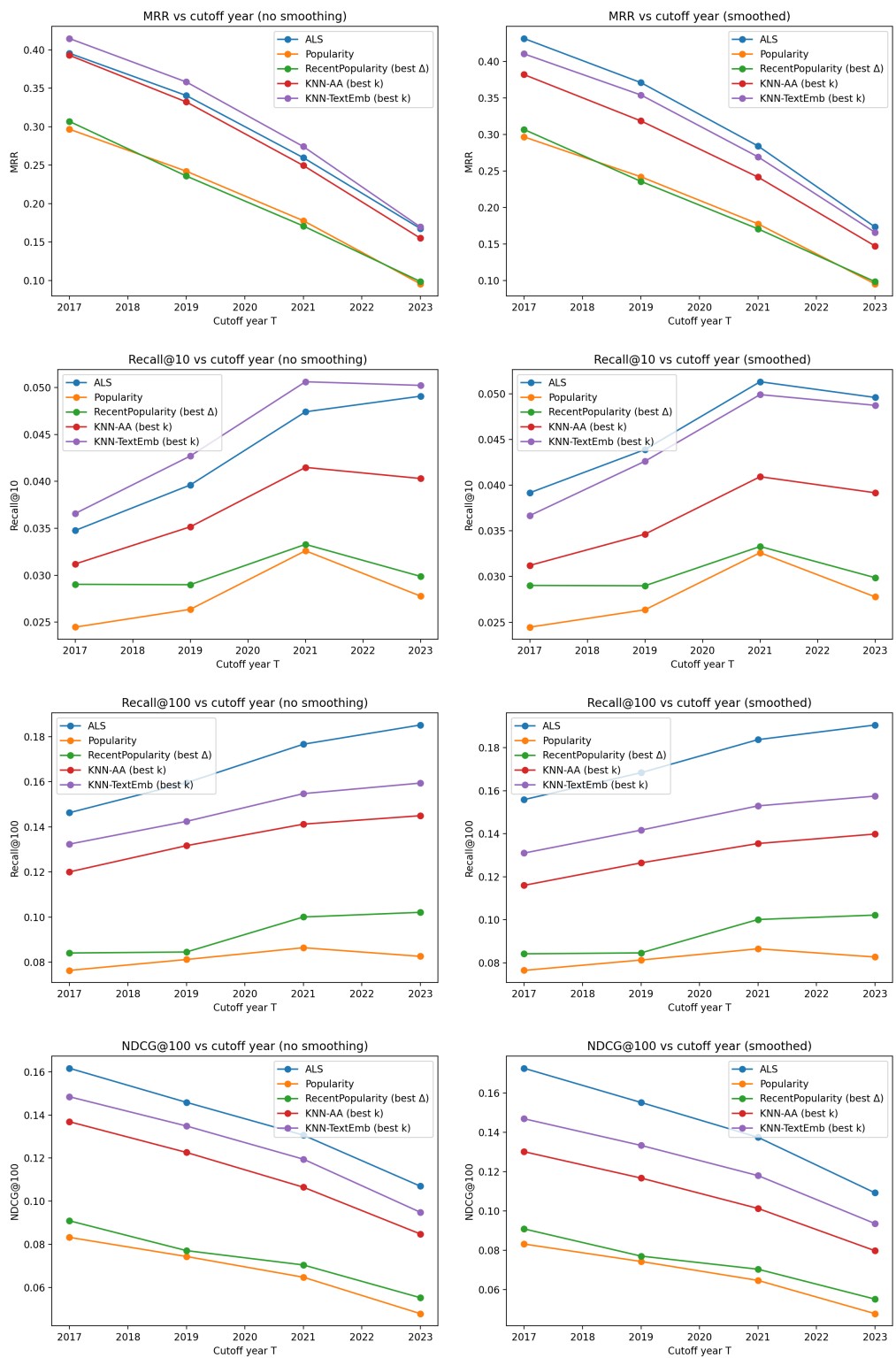

Figure 5: Metric trends across cutoff years (Physical subset). Each row shows one metric (top to bottom: MRR, Recall@10, Recall@100, NDCG@100); **left column:** without smoothing, **right column:** with smoothing. Lines correspond to ALS (blue), Popularity (orange), RecentPopularity with best $\Delta$ (green), ConceptKNN-AA with best $k$ (red), and ConceptKNN-TextEmb with best $k$ (purple). Smoothing uniformly improves all methods; with smoothing, ALS leads on all metrics at every cutoff.

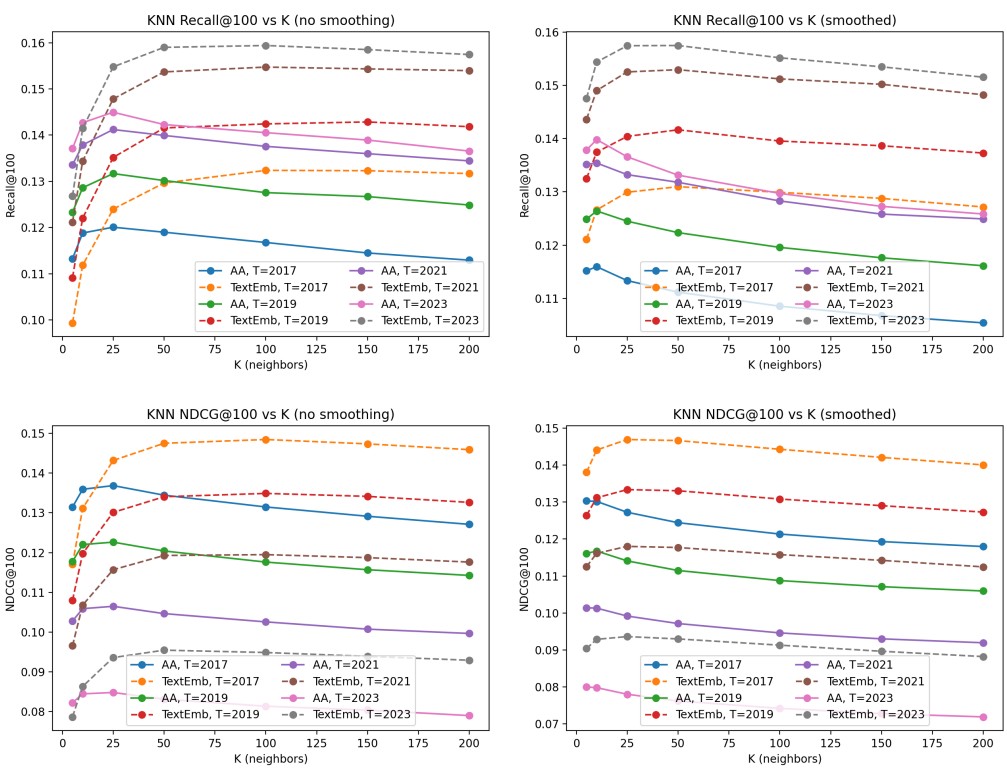

Figure 6: KNN performance vs. neighborhood size $k$ (Physical subset). Each row shows one metric (top: Recall@100, bottom: NDCG@100); **left column:** without smoothing, **right column:** with smoothing. Solid lines denote ConceptKNN-AA and dashed lines denote ConceptKNN-TextEmb; colors distinguish cutoff years ($T \in \{2017, 2019, 2021, 2023\}$), with later cutoffs generally achieving higher scores. Performance is stable for $k \gtrsim 50$; smoothing shifts curves upward without changing their dependence on $k$.

