# OpenReview forum: "Predicting New Concept--Object Associations in Astronomy by Mining the Literature"
_ICLR.cc/2026/Workshop/FM4Science — ICLR 2026 Workshop FM4Science Poster_

### Official Review · Reviewer_ii2T · 2026-02-22
**Concept-Object associations is an interesting approach for literature-driven research. As a POC, this is an acceptable work. However, I think there are significant questions about the relevance of such method in real research.**

**Rating:** 6
**Confidence:** 3

**Review:**

The authors presented a new approach for the prediction of future concept-object associations based on historical relationships. They construct a knowledge graph with concept-object connections, based on historical papers, and optimize learned concept and object embeddings using the ALS method. This is a timely work as it correctly identifies the conceptual gap between astrophysical datasets and literature. The authors mitigate semantic overlap by smoothing the model's base score over neighbouring concepts. While the approach is convincing, I see several limitations. My main concern is that using associations that are based on appearance in preprint papers can sometimes be misleading. While the authors correctly noted that not all mentions are equal and give weight to each mention based on its role in the paper and its study mode, I think there are many more 'failure modes'. One example is misclassification - sometimes objects are classified wrongly (for example, a companion star in a binary system can be identified as a planet), and a large amount of the scientific work is about rejecting hypotheses, namely showing that objects are not what they were thought to be. The suggested method not only misses this type of work, but it can also harm it, creating false historical continuity. Another problem is the mixture of systematics and physics concepts. The authors partially mitigated this by excluding non-physical concepts, but I argue that the mixture is much more nuanced, as many surveys have physical limitations (for example, most surveys are magnitude-limited, which creates implicit bias for certain types of objects). In addition, papers can have (and usually have) mistakes, and the fact that the authors used preprints, where at least some of them have not gone through a peer review process, makes them more prone to such mistakes. This might be mitigated, at least partially, by taking into account the citations of a paper. Lastly, if I understand correctly, the predictions of the model are 'point estimates', without any confidence interval. As the authors correctly frame the model as 'a tool to inform judgment, not to replace it', and given all the potential biases of such a method, uncertainty estimation is crucial.
To conclude, I think this work suggests an interesting approach to connecting literature and astrophysical datasets. However, there are major concerns about the effectiveness and biases of such a method. Some issues are hard to mitigate (for example, physics-sytematic coupling), but at least should be acknowledged.

---

### Official Review · Reviewer_SxMm · 2026-02-23
**A meaningful and relevant contribution, albeit with some technical limitations.**

**Rating:** 5
**Confidence:** 4

**Review:**

This paper introduces a large-scale dataset and empirical study for concept–object forecasting in astronomy, constructed by mining the scientific literature and modeling temporal concept–object associations. The work frames the task as a form of literature-based discovery and evaluates several matrix factorization–based baselines under temporal splits.

**strength**
1. The paper addresses scientific discovery through literature mining and forecasting, and the astronomy use case is well aligned with FM4Science.
2. The concept–object forecasting setup is clearly described, with appropriate temporal splits and multiple evaluation metrics across different cutoffs.

**wearkness**
1. The first part of the introduction focuses broadly on scientific discovery and link prediction, while the astronomy-specific motivation and scientific constraints (e.g., telescope time scarcity) are introduced later. Anchoring the concrete scientific use case earlier would improve clarity and framing.
2. Several figures emphasize conceptual illustration over data-driven analysis. While useful for overview, they provide limited direct support for the empirical claims and make it harder to assess the structure and scale of the constructed graph.
3. The modeling approach relies on relatively standard matrix factorization techniques with smoothing. The methodological contribution is modest.
4. The paper constructs 9,999 scientific concepts via k-means clustering over LLM-derived embeddings. While practical, hard clustering introduces semantic boundary effects by forcing overlapping concepts into mutually exclusive clusters, potentially fragmenting coherent scientific associations.

---

### Official Review · Reviewer_3E3s · 2026-02-24
**Very meaningful information retrieval system for astronomy**

**Rating:** 8
**Confidence:** 4

**Review:**

Quality: 8/10
High-quality work overall. Strong problem and solution design. Careful temporal evaluation. Good baselines. Honest limitations section
Clarity: 8.5/10
The paper is clear for a domain knowledge heavy ML-for-science paper.
Originality: 7/10
the combination in this astronomy setting is original and valuable: first large-scale concept–object graph in astronomy (their claim). explicit forecasting formulation tied to telescope triage. concept-overlap smoothing for clustered concept vocabularies
Significance: 8/10
defined a very valuable question, and bringing real values for astronomy researchers

pros:
1. meaningful real research problem
2. solid solution

cons:
1. in the future: can consider deep learning for matching or more modern LLM information retrieval method, including embedding retrieval, LLM ranker. and keep more raw-text content with the object / literature, do it in a more E2E modern machine learning way.

---

### Meta-Review · Area_Chair_MfRV · 2026-02-27

**Recommendation:** Accept (Poster)
**Confidence:** 4

**Metareview:**

This submission has received three reviews. Two positive reviews with a "clear accept" and a "marginally above acceptance threshold". One reviewer rated the paper with a "marginally below acceptance threshold".

After reading the reviews, I recommend this paper for "acceptance" and ask the authors to consider implementing the feedback given by all reviewers into the camera-ready version of the paper.

---

### Decision · Program_Chairs · 2026-03-03

Accept (Poster)